# Essential Fatty Acid Deficiency in Cystic Fibrosis Disease Progression: Role of Genotype and Sex

**DOI:** 10.3390/nu14214666

**Published:** 2022-11-04

**Authors:** Nirajan Shrestha, Alexandra McCarron, Nathan Rout-Pitt, Martin Donnelley, David W. Parsons, Deanne H. Hryciw

**Affiliations:** 1School of Pharmacy and Medical Sciences, Griffith University, Southport, QLD 4222, Australia; 2Adelaide Medical School, University of Adelaide, Adelaide, SA 5001, Australia; 3Robinson Research Institute, University of Adelaide, Adelaide, SA 5001, Australia; 4Respiratory and Sleep Medicine, Women’s and Children’s Hospital, 72 King William Road, North Adelaide, SA 5006, Australia; 5School of Environment and Science, Griffith University, Nathan, QLD 4111, Australia; 6Institute for Health and Sport, Victoria University, Melbourne, VIC 3000, Australia; 7Griffith Institute for Drug Discovery, Griffith University, Nathan, QLD 4111, Australia

**Keywords:** cystic fibrosis, essential fatty acid, sex, genotype

## Abstract

Adequate intake of nutrients such as essential fatty acids (EFA) are critical in cystic fibrosis (CF). The clinical course of deterioration of lung function in people with CF has been shown to relate to nutrition. Independent of the higher energy consumption and malabsorption due to pancreatic insufficiency, EFA deficiency is closely associated with the risk of pulmonary infection, the most significant pathology in CF. This review will focus on the EFA deficiency identified in people with CF, as well as the limited progress made in deciphering the exact metabolic pathways that are dysfunctional in CF. Specifically, people with CF are deficient in linoleic acid, an omega 6 fatty acid, and the ratio of arachidonic acid (omega 6 metabolite) and docosahexaenoic acid (omega 3 metabolite) is increased. Analysis of the molecular pathways in bronchial cells has identified changes in the enzymes that metabolise EFA. However, fatty acid metabolism primarily occurs in the liver, with EFA metabolism in CF liver not yet investigated, indicating that further research is required. Despite limited understanding in this area, it is well known that adequate EFA concentrations are critical to normal membrane structure and function, and thus are important to consider in disease processes. Novel insights into the relationship between CF genotype and EFA phenotype will be discussed, in addition to sex differences in EFA concentrations in people with CF. Collectively, investigating the specific effects of genotype and sex on fatty acid metabolism may provide support for the management of people with CF via personalised genotype- and sex-specific nutritional therapies.

## 1. Introduction

Cystic fibrosis (CF) is an inherited, autosomal recessive disorder, caused by pathogenic variants of the CF transmembrane conductance regulator (*CFTR*) gene [1]. The CFTR protein is a chloride ion channel expressed in epithelial cells, and six classes of CFTR mutations can result: a complete absence of the CFTR protein (Class I); trafficking mutations leading to CFTR processing deficiencies that result in misfolding and premature degradation by the endoplasmic reticulum (Class II); gating mutations that cause a loss of CFTR channel function (Class III); reduced conductance of the CFTR protein (Class IV); mutations that significantly reduce the amount of CFTR protein at the cell surface (Class V); and mutations that make cell surface CFTR less stable (Class VI) [2]. The most common mutation is a homozygous loss of the phenylalanine residue at position 508 of the CFTR protein (Phe508del), occurring in ~70% of people with CF, with 85.8% of individuals harbouring this mutation on one allele (https://www.cff.org/sites/default/files/2021-11/Patient-Registry-Annual-Data-Report.pdf, accessed on 12 February 2022).

CF affects multiple organs, including the lung, gut, pancreas, and reproductive tract, with its impact on the respiratory system resulting in the greatest morbidity and mortality [3]. People with CF typically have recurrent pulmonary infections, which over time become chronic, leading to fibrosis, pulmonary failure, and death [1]. In addition to poor clearance of bacteria, lung disease in people with CF is partly attributed to an excessive inflammatory response. CF lung inflammation is characterised by increased neutrophil infiltration and production of pro-inflammatory cytokines [4]. Strikingly, infants with CF demonstrate lung inflammation and structural changes such as bronchiectasis within the first few weeks of life while they are asymptomatic and culture-negative for CF-relevant pathogens [5]. Moreover, while infection is known to exacerbate lung inflammation, studies using CF human bronchoalveolar lavage samples indicate that inflammation is relatively independent of past or current infection status, suggesting other factors contribute to this state of hyperinflammation [5]. Notably, the presence of lung inflammation in infants with CF has been associated with a poorer nutritional status, including a reduced body mass index [6]. The immune response can be regulated by the ratio of omega 6 and omega 3 essential fatty acids (EFA) [7], indicating that nutrition may be critical in the development of CF-related pathologies.

Adequate nutrition is critical for people with CF and nutritional status plays an important role in the progression of the pulmonary disease. Certainly, dysfunctional lipid metabolism is a hallmark indicator of the disease [8]. CF is associated with a higher energy consumption as a consequence of nutritional deficiencies and malabsorption associated with pancreatic insufficiency [9]. Improvements in nutritional status in people with CF positively impacts morbidity and mortality [10]. EFA deficiency in CF is common, with emerging research suggesting the CF genotype and sex play a role in its severity. This review aims to provide a comprehensive overview of the role of EFA deficiency in CF disease progression, with reference to genotype and sex.

## 2. Metabolism of Dietary Polyunsaturated Fats

Polyunsaturated fatty acids (PUFAs) play an important part in the structure and function of cellular membranes and are precursors for lipid mediators, which play a key role in maintaining normal cellular functions. During digestion, PUFAs are released from dietary triglycerides as free fatty acids and absorbed in the small intestine via specific transmembrane (fatty acid transport (FATP)) and intracellular (fatty acid binding (FABP)) proteins. In the intestine, fatty acids are resynthesized into triglycerides and secreted into the plasma. In the liver they are stored or released as required [11]. In the intestine, fatty acids are taken up by FATP4 [12] and intracellularly by FABP 1/2/6 in the small intestine [13], where they are activated by acyl-CoA, then shuttled via acyl-CoA binding protein to mitochondria or peroxisomes for β-oxidation, or to the endoplasmic reticulum for esterification. Some free fatty acids may bind to transcription factors that regulate gene expression or may be converted into signalling molecules (eicosanoids).

EFAs cannot be synthesised in humans and mammals, and as such, can only be obtained from the diet. In humans, the only two EFAs are linoleic acid (LA), an omega 6 PUFA, and alpha-linolenic acid (ALA), an omega-3 PUFA. Once ingested, LA and ALA are metabolised by the same desaturation and elongation enzymes, specifically fatty acid desaturase (FADS) 1 and FADS2 [14]. Elevated consumption of LA leads to a dampening of the metabolism of ALA and increased metabolism of LA [15]. LA and ALA are metabolised into arachidonic acid (AA) and docosahexaenoic acid (DHA), respectively. AA is proinflammatory, while DHA is anti-inflammatory. The balance between AA and DHA is critical in regulating inflammatory processes and impacting the development and severity of inflammatory diseases, including atherosclerosis and other cardiovascular diseases [16]. Excess AA is proinflammatory as it leads to the production of prostaglandin (PGE2) and leukotriene (LTB4), and a diet rich in LA increases the risk of inflammatory changes in pathological pathways in a number of organ systems, including in the lungs [17]. 

## 3. Fatty Acid Abnormalities in Cystic Fibrosis

People with CF have historically been advised to eat diets consisting of high-calorie, high-fat (40%), high-salt foods, along with fruits and vegetables (https://cystic-fibrosis.com/diet-nutrition/, accessed on 12 February 2022). The imbalance in EFAs was identified in people with CF before the genetic defect associated with the disease was identified [8]. Many studies have shown that despite greater energy, saturated fat, and LA intake, people with CF have reduced LA concentrations in the serum [18,19], tissues [20], and erythrocytes [21]. Further, in the serum of people with CF, there is a lower DHA and a higher AA:DHA ratio [22], and a higher dihomogammalinolenic acid, oleic acid, and mead acid serum concentration, resulting in an increased ratio of mead acid/AA and oleic acid/LA [23]. 

Research using bronchial cell culture models has attempted to identify the molecular pathways involved in altered EFA metabolism in CF. In 16HBEo^−^ CF epithelial cells where *CFTR* is silenced, FADS1 and FADS2 expression are increased, in addition to the cyclooxygenase (COX) COX2 [24]. Further, in control cells, ALA is the preferred substrate for FADS2, with a two to three times higher affinity than LA [25]. This would lead to an increased amount of AA and DHA in CF; however, typically an increase in AA and reduction in DHA are observed, suggesting a higher affinity for LA in CF. As LA is typically higher in the diet, it suggests that in CF there is a higher rate of metabolism of LA over ALA. To add to this, in CF there is elevated expression of AMP-activated protein kinase (AMPK), which drives upregulation of FADS1/2 [26]. 

It is important to note that serum and cellular concentrations of EFA can differ (e.g., [27]). For example, in a small cohort, lung transplantation in people with CF restored plasma LA concentrations, but did not alter the AA:DHA ratio [28]. More recent research has demonstrated that lung transplantation in people with CF improved the total omega 3, EPA, DHA, and the omega 6 to omega 3 ratio in the plasma, but not the erythrocyte membrane [27]. Therefore, differences in the EFA profiles in people with CF between studies have been observed. Witters et al. [28] proposed that there is normalisation of serum EFA due to restored CFTR function in the lung in people with CF who have had lung transplantation. Hanssens et al. [27] proposed that a lung transplantation improved inflammation in people with CF, due to attenuation of the excessive EFA oxidation and increased LA to AA metabolism. This may be in part due to the presence of functional CFTR and the effects of immunosuppressive therapy associated with the transplantation. Notably, the liver is the major organ for fatty acid metabolism [11], and changes in the pathways involved in fatty acid metabolism in the liver have not been investigated. 

LA is an integral component of cell membranes and the ratio of omega 6:omega 3 and cholesterol is important for transmembrane insertion and function [29]. There have been several investigations to identify the underlying mechanism linking EFA deficiency and CF, as pancreatic enzyme replacement therapy does not restore the deficiency [30]. In addition, people with CF are documented to have higher oxygen consumption due to their increased basal metabolic rate [31]. The consequence of this increased energy requirement could be increased oxidation of LA and other fatty acid to meet energy needs, which would serve to decrease the availability of LA for use as an EFA [31]. Finally, there is altered expression of enzymes that metabolise EFA, which may contribute to the imbalance of the AA:DHA ratio. For example, phospholipase A2 (PLA2) releases AA from cell membranes to increase the circulating AA concentrations; in turn, COX1 and COX2 metabolise AA to produce prostaglandins [32]. Notably, in Cftr-deficient mice, and people with CF, cytosolic PLA2α activity is increased [33], and COX1 and COX2 mRNA and protein expressions are increased in nasal polyps from people with CF [34]. 

Studies have shown a link between low DHA in CF-related liver disease [35]. However, use of DHA supplementation as a potential therapeutic is not well supported as it does not suppress the inflammatory profile of people with CF [36]. More broadly, the use of omega 3 supplementation in improving the outcomes of people with CF is not well supported by clinical studies [37]. This may be in part due to the cellular effects of the CFTR mutations, which may create a localized proinflammatory lipid environment due to elevated concentrations of AA.

Interestingly, a recent review has suggested the relationship between CFTR function and fatty acids is bidirectional [38]. In vitro, expression of Phe508del-CFTR reduces incorporation of LA into the plasma membrane [39]. This may be because active incorporation of fatty acids in the plasma membrane is influenced by chloride channels [39]. In people with CF, expression of Phe508del-CFTR, which is trapped in the endoplasmic reticulum, causes a redistribution of free cholesterol, with perturbations in cholesterol trafficking in the cell [40]. Specifically, cholesterol accumulates in the endosomal/lysosomal compartments in cells expressing Phe508del-CFTR [40]. This block in translocation of cholesterol to the Golgi and endoplasmic reticulum for esterification may contribute to the compromised trafficking of the Phe508del-CFTR (Figure 1).

The mechanism for this is not fully understood, but due to LAs critical role in plasma membrane structure [42], it may be the case that low LA contributes to the cholesterol redistribution and supplementation of LA may increase cell surface expression of Phe508del-CFTR. Based on this, when the CFTR protein is absent (e.g., in Class I CFTR mutations), the lack of CFTR chloride current means that LA/DHA supplementation will not increase LA in the membrane. When Phe508del-CFTR is expressed, a high endoplasmic reticulum cholesterol contributes to the defect in trafficking (Class 2 mutation). Elevated LA/DHA may cause translocation of the cholesterol and increase CFTR in the plasma membrane, which, in turn, will increase LA in the plasma membrane.

## 4. Effect of Genotype

It has long been established that the severity of the CF clinical syndrome varies due to a range of factors, including genotype [43]. Prior to the advent of newborn screening, people with CF were diagnosed anywhere between birth and adulthood, depending on their disease severity. In addition, there is phenotypic variability in people with CF with the same *CFTR* genotype [44]. Those homozygous for Phe508del typically have a severe form of the disease, with an earlier onset, elevated sweat chloride, and pancreatic insufficiency [43].

Limited studies have investigated the nutritional status of people with CF with respect to their genotype. In one study, CF people homozygous for Phe508del or heterozygous/homozygous for 394delTT showed significantly lower concentrations of LA and DHA compared to people with CF who had a rare CFTR mutation [45]. Interestingly, people with mutations in Classes I, II, and III have lower LA and a higher DHGLA (dihomo-gamma-linolenic acid), 22:4 omega-6, 22:5 omega-6, and 20:3 omega-9 [23]. This suggests that the severity of the CF phenotype relates to the EFA status and mutation class. At this time, the underlying mechanism is not known; however, it partly could be due to the link between EFA status and chloride transport through the membrane (as described in Figure 1).

In another study, Medza et al. compared people with CF who were either heterozygous or homozygous for Phe508del [46]. They demonstrated that the serum vitamins and erythrocyte membrane fatty acid profiles were the same in both groups, except for heptadecanoic acid. Further, the mean percentiles of height, weight, and body mass index did not differ significantly between the two groups, suggesting their nutritional statuses were similar. Of interest, in two mouse models of CF (homozygous Phe508del and Cftr knockout) there were no differences in EFA status compared to wildtype mice, specifically LA, ALA, AA, and DHA concentrations in the pancreas, lung, and jejunum [47]. Of note, in a different *CFTR* knockout mouse model, a membrane lipid imbalance was found in the ileum, pancreas, and lung, with an increase in phospholipid-bound AA and a decrease in phospholipid-bound DHA [48]. These results suggest that the background mouse strain may cause variation in fatty acid metabolism.

Part of the variability in the CF clinical syndrome may be due to mutations and polymorphisms in genes that are critical to EFA metabolism. A study proposed a link between variants in the COX1 and COX2 enzymes and CF outcomes in homozygous Phe508del people [49]. Specifically, 639C>A and 762 + 14delA polymorphisms of the COX1 gene reduced COX1 protein expression and were associated with an improvement in lung function. Further, the -765G>C and 8473T>C polymorphisms decreased COX2 expression and improved lung function and reduced *Pseudomonas aeruginosa* infections [49]. Collectively, these variants demonstrate the impact COX1 and COX2 have on CF pathology. Notably, inhibition of COX2 independent of COX1 increases the risk of serious cardiovascular events [50]. Despite variants in FADS1/2 leading to dysfunctional EFA metabolism [51], at this time, the link between FADS1/2 polymorphisms and CF severity is unknown.

CFTR modulator therapeutics that provide clinical respiratory benefits to people with CF have recently become widely available. These therapeutics increase the number of CFTR channels and increase CFTR channel activity, restoring chloride transport and reversing the primary defect in people with CF [52]. The effects of modulator therapies on EFA profiles have had limited investigation. In people with the G551D CFTR mutation, treatment with the modulator Ivacaftor did not restore LA or DHA but did decrease AA [53]. Some more recently developed therapeutics are Elexacaftor, Tezacaftor, and Ivacaftor (marketed as Trikafta or Kaftrio), which target Phe508del homozygous and Phe508del heterozygous mutations in CFTR [54]. Trikafta improves lung function in people with CF [55]. At this time, the effects of Trikafta therapy on EFA status in people with CF is unknown. Thus, further investigation into the effects of potential modulators of CFTR function on EFA metabolism as therapeutic targets for the restoration of EFA concentrations in people with CF is warranted.

## 5. Effect of Sex

It was established a number of years ago that, among people with CF between the ages of 1 to 20 years, females have a higher mortality rate than males [56]. Specifically, this research demonstrated that females with the same CF genotype have an increased morbidity, and mortality occurs earlier compared to males. Mechanistically, there is increasing evidence that the observed sexual dimorphism is due to estrogen, which exacerbates decline in lung function, infection, and inflammation in females with CF—the so-called “CF gender gap” [57]. In the lungs, a major cause of pulmonary infections in people with CF is *P. aeruginosa.* Estrogen exacerbates the *P. aeruginosa* virulence factors and enhances the interaction between the bacterium and the bronchial epithelium [57]. Further, in females with CF, use of the oral contraceptive pill, which decreases endogenous estrogen concentrations and increases synthetic oestradiol concentrations, reduces the requirement for antibiotics to treat the ‘mucoid’ form of *P. aeruginosa* [58]. This suggests that high estrogen may modulate the microbiome. In the context of controlling EFA concentrations, this suggests that estrogen-mediated microbiome diversity [59] may play a role in EFA status in people with CF.

Limited research has investigated the differences in fatty acids in circulation and in the erythrocyte membranes as it relates to sex [60]. Coste et al. [60] identified a significantly lower DHA and higher eicosatrienoic acid in plasma and erythrocyte membranes in males with CF compared with females with CF. However, in general, in people without CF, males have a lower concentration of AA and DHA in plasma and in plasma phospholipids compared to women [61]. Further, plasma ALA concentrations are significantly reduced in males with CF compared with females with CF [60]. At this time, what remains unknown are the changes in the molecular pathways responsible for metabolism of fatty acids that lead to these differences between the sexes. Though not mainstream in the field, it is critical that analyses of pathological changes, and in particular changes to EFA status in people with CF, are separated based on sex.

## 6. Concluding Remarks

In summary, this review discusses the emerging links between genotype and EFA concentrations, as well as sex and EFA concentrations in people with CF. As the underlying mechanism for EFA deficiency in CF is unknown, future investigations should have genotype and sex as confounders for data analysis associated with EFA status and metabolism in people with CF. Understanding genotype and sex differences are critical to develop tailored and personalised solutions to restoring EFA status in a clinical context.

## Figures and Tables

**Figure 1 nutrients-14-04666-f001:**
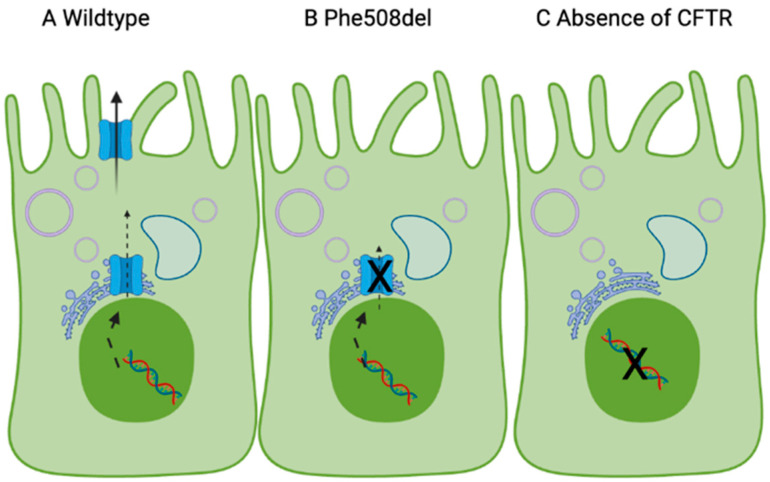
Proposed model linking CFTR expression and essential fatty acids in the cell membrane, based on [39,41]. The CFTR gene (nucleus) codes for the CFTR protein, which is made in the endoplasmic reticulum and trafficked to the apical membrane. In (**A**), wildtype CFTR leads to incorporation of LA and DHA in the cell membrane and normal chloride transport. (**B**) Phe508del-CFTR and (**C**) absence of CFTR at the cell surface reduces incorporation of LA/DHA in the plasma membrane due to dysfunctional chloride transport. Made with Biorender.

## Data Availability

Not applicable.

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
