# Peer review of "Essential Fatty Acid Deficiency in Cystic Fibrosis Disease Progression: Role of Genotype and Sex"

_nutrients, 2022, doi:10.3390/nu14214666_

Round 1
Reviewer 1 Report
This review discusses essential fatty acids (EFAs) in CF. It describes the deficiency in linoleic acid and abnormally high arachidonic acid to docosahexaenoic acid ratio. The enzymes that metabolize EFAs are briefly described as are the roles of EFAs in membrane structure and function. Hypotheses to explain the relationship between CF genotype and EFA phenotype are discussed, especially sex differences in EFA concentrations in CF patients. Overall it provides a good overview of this poorly understood topic.
This is a topical review that highlights specific aspects rather than a comprehensive review of the literature, thus many references demonstrating the EFA imbalance are missing. However a topical review seems appropriate at this stage when a clear understanding of EFAs in CF is not yet available in the primary literature and mechanisms that have been proposed recently in other reviews, such as the one cited by McCarty and colleagues and reiterated here, are highly speculative. The manuscript is clear and well written overall. The most novel part comes is the interesting discussion of sex differences.
The main weakness is that it conflates availability of EFAs with the EFA imbalance that occurs at the cellular level. The high AA to DHA ratio is a cell-autonomous defect which arises when CF and non-CF cells are cultured in vitro under identical conditions (i.e. in the same medium). That finding indicates that EFA availability is not an essential determinant of the lipid imbalance that characterizes CF. That does not mean we should ignore the nutritional EFA supply or the liver, however it indicates that CFTR mutations acting at the cellular level probably create the proinflammatory lipid environment in airway cells independently of dysfunction in the GI tract and liver. That is probably why omega 3 supplementation has not worked therapeutically. This manuscript clarify the systemic vs cell-autonomous EFA imbalance in CF and explain how they may be linked.
Reviewer 2 Report
I commend the authors for their detailed, informative, and excellent review of this important topic – the role of EFA in CF.
While I see no major concerns, the following minor suggestions will make the paper more interesting.
1. Title – consider changing to “Essential Fatty acid deficiency in cystic fibrosis disease progression: Role of genotype and sex”
2. Line 122 – Please add this study on lung transplant patients CF as well (Hanssens L, Duchateau J, Namane SA, Malfroot A, Knoop C, Casimir G. Influence of lung transplantation on the essential fatty acid profile in cystic fibrosis. Prostaglandins, Leukotrienes and Essential Fatty Acids. 2020 Jul 1;158:102060.) This study highlighted that lung transplantation improved the EFA profile in the plasma but not in the erythrocyte membrane.
3. Effect of Genotype – In this section, add a brief discussion on the effect of CFTR-directed modulator therapies on EFAs (O'Connor MG, Seegmiller A. The effects of ivacaftor on CF fatty acid metabolism: An analysis from the GOAL study. Journal of Cystic Fibrosis. 2017 Jan 1;16(1):132-8.) As we are currently in the era of highly effective modulator therapies (Ivacaftor, and triple comb – Elexacaftor/Tezacaftor/Ivacaftor), the impact of these therapies on EFA metabolism is very important.
